# Mesenchymal Stem/Stromal Cells and Their Paracrine Activity—Immunomodulation Mechanisms and How to Influence the Therapeutic Potential

**DOI:** 10.3390/pharmaceutics14020381

**Published:** 2022-02-09

**Authors:** Rui Alvites, Mariana Branquinho, Ana C. Sousa, Bruna Lopes, Patrícia Sousa, Ana Colette Maurício

**Affiliations:** 1Centro de Estudos de Ciência Animal (CECA), Instituto de Ciências, Tecnologias e Agroambiente (ICETA) da Universidade do Porto, Praça Gomes Teixeira, Apartado 55142, 4051-401 Porto, Portugal; rmalvites@icbas.up.pt (R.A.); mevieira@icbas.up.pt (M.B.); up201502681@up.pt (A.C.S.); bilopes@icbas.up.pt (B.L.); 201403005@icbas.up.pt (P.S.); 2Departamento de Clínicas Veterinárias, Instituto de Ciências Biomédicas Abel Salazar (ICBAS), Universidade do Porto (UP), Rua de Jorge Viterbo Ferreira, nº 228, 4050-313 Porto, Portugal

**Keywords:** Mesenchymal Stem/Stromal Cells, secretome, extracellular vesicles, homing, immunomodulation, regenerative medicine

## Abstract

With high clinical interest to be applied in regenerative medicine, Mesenchymal Stem/Stromal Cells have been widely studied due to their multipotency, wide distribution, and relative ease of isolation and expansion in vitro. Their remarkable biological characteristics and high immunomodulatory influence have opened doors to the application of MSCs in many clinical settings. The therapeutic influence of these cells and the interaction with the immune system seems to occur both directly and through a paracrine route, with the production and secretion of soluble factors and extracellular vesicles. The complex mechanisms through which this influence takes place is not fully understood, but several functional manipulation techniques, such as cell engineering, priming, and preconditioning, have been developed. In this review, the knowledge about the immunoregulatory and immunomodulatory capacity of MSCs and their secretion products is revisited, with a special focus on the phenomena of migration and homing, direct cell action and paracrine activity. The techniques for homing improvement, cell modulation and conditioning prior to the application of paracrine factors were also explored. Finally, multiple assays where different approaches were applied with varying success were used as examples to justify their exploration.

## 1. Introduction

Mesenchymal Stem Cells (MSCs) are a category of multipotent, non-hematopoietic cells with mesodermal origin. Additionally to their typical fibroblast-like morphology when in culture, MSCs are also able to differentiate into typical cells of the mesodermal (such as adipocytes, chondrocytes and osteocytes), endodermal (alveolar endothelial cells) and neurectodermal (neuroglial cells) lineages [1].

Their self-renewable nature and relative ease of in vitro expansion have made MSCs one of the pillars for modern cell-based therapies applied to promote tissue regeneration [2]. Since the identification in bone marrow [3], MSCs have been detected and effectively isolated from nearly all tissues. Dental pulp, umbilical cord and placental tissues, neonatal tissues, adipose tissue, peripheral blood, skin, and olfactory mucosa are among the most widely studied niches where MSCs can be isolated and mobilized from, but the list is long and is constantly being updated [4]. Despite their ubiquity in the body, the functional characteristics of MSCs vary depending on the niche. In addition to the practical limitations associated with the collection of some tissues by invasive techniques, different origins of MSCs translate into different proliferation rates, in vitro differentiation capacity, and ranges of application. Thus, the selection of a type of MSC to be applied in a given clinical or pre-clinical setting should be based on a balance of all logistical and practical variables, as well as on the functional behavior of the cells in culture [5].

To standardize the scientific debate, the International Society for Cellular Therapy (ISCT) established the minimum criteria that must be identified in these cells so that they can be classified as MSCs, namely adherence to plastic surfaces when in culture, the ability to differentiate into three mesodermal cell lineages and the expression of certain cell surface antigens. For the specific case of human cells where these criteria were initially established, MSCs (hMSCs) must be positive for markers such as CD73, CD90 and CD105, and negative for other markers such as CD11b, CD14, CD19, CD34, CD45, CD79 and HLA- DR (Table 1) [6]. These restrictive criteria quickly became obsolete, and today several other markers such as CD271 and Stro-1 are used as identifiers of MSCs [7,8], cells whose ability to differentiate into lineages other than the mesodermal has been repeatedly proven. Animal-derived MSCs (aMSCs) also poorly fit the original criteria, but despite its practical limitation, the ISCT has never updated the parameters, which continue to be applied to both hMSCs and aMSCs [9].

In the early days of their application in vivo, it was noticed that, once in the systemic circulation, MSCs had the ability to migrate (or homing) to sites with lesions and there replace the injured resident cells directly or after differentiation, thus promoting tissue regeneration and acting as a therapeutic agent [11]. It was later noticed that, in addition to their direct pro-regenerative capacity, these cells also act through a paracrine action and cell-to-cell signaling/communication, secreting cytokines and growth factors [12]. Firstly, MSCs migrate to sites of injury attracted by chemotactic signals released by damaged tissues. Through the secretion of soluble factors, they manage to control cell viability and proliferation and modulate directly or indirectly the local and systemic immune response, by interacting with both the myeloid and lymphoid cells of the innate and adaptive immune system. Although the mechanisms through which MSCs modulate and modify the immune response remains a mystery, it is now clear that the release of soluble factors, extracellular vesicles and exosomes is behind their immunomodulatory action [13].

The desire to apply MSCs in a context of clinical translation created the need to more actively explore the mechanisms of action of these cells, thus allowing determination of the safety and feasibility of their application in humans and veterinary species for the treatment of different pathologies [9,14]. From an initial dubious characterization, due to the demonstration that their clinical application can improve the functionality of damaged tissues and organs and is not associated with harmful effects, MSCs have recently started to be included into the category of medicines [15]. The two main western medicine agencies saw the need to integrate these therapeutic products from a legal point of view, establishing a specific legal framework and a well-defined jurisdiction. The European Medicine Agency (EMA) integrates MSCs and their derivative products into the categories of Advanced Therapies Medicinal Products (ATMPs) [16], and their use is regulated by Regulation No. 1394/2007 [17]. The Food and Drug Administration (FDA) categorizes these products as Human Cellular and Tissue-based Products (HCT/Ps) [18], regulating their application through the Code of Federal Regulations, part 1271(21 CFR 1271) as biologicals [19].

## 2. Mesenchymal Stem Cells (MSCs) Immune Action Mechanisms

### 2.1. Delivery and Mobilization

One of the advantages of applying MSCs therapeutically is their ability to migrate to damaged tissues. This migration or homing depends largely on the origin of the cells and on whether they have an exogenous origin (injected) or if they act endogenously from the body’s organic reservoirs (niches). The exogenous homing process can either be systemic or non-systemic. In non-systemic homing, cells are transplanted directly into the injured tissue, and then migrate to the specific lesion site attracted by the gradient of chemokines [20]. In the case of systemic mobilization, MSCs follow a sequence of phases that allow them to migrate into the injured tissue and site, namely by adhesion, activation, entrapment, diapedesis, and migration phases.

Direct administration of MSCs is the fastest way to induce therapeutic effects at the site and to promote regeneration. Cells administered using this pathway can more easily follow homing and differentiate into the type of cells and tissues needed through self-renewal processes. Secretion products such as exosomes can also be administered by this route. The main advantage over systemic administration is the smaller dilution and loss of cells and secreted bioactive factors by degradation in the bloodstream [21]. Examples of direct administration with proven effects include intrathecal or intraspinal administrations for spinal cord injuries [22], intra-articular for osteoarthritis and cartilage repair [23,24], intramuscular for volumetric skeletal muscle injuries [25], direct transplantation for resolution of peripheral nerve injuries [26] and local administration for healing of bone fractures [27]. However, there are still many doubts regarding the best method for direct administration of MSCs in the injured tissue, and the migration of the cells or their products via bloodstream or lymphatic drainage can significantly decrease the number or concentration of cells and reduce their action on the site and neighboring tissues. The number of cells needed for adequate local efficacy may require more complex surgical interventions, and the non-interaction with secondary signaling systems at the systemic level may limit the full range of action of MSCs. Furthermore, in more severe cases, intense local inflammatory processes can create a hostile environment that is not favorable to the survival and action of the administered cells [21].

Systemic administration, mostly intravenous, has the obvious advantages of granting a rapid systemic delivery, allowing MSCs to reach all injured tissues attracted by the gradient of inflammatory cytokines [28]. However, disadvantages of this route also exist, namely with a marked dilution of cells in the bloodstream and retention in the pulmonary vasculature, with a lower concentration of MSCs or secretion products reaching the sites to be treated [21]. In some specific cases, such as the administration of MSCs for the treatment of neurological diseases, there is a risk of cellular emboli or thrombi development in the cerebral microvasculature, with exacerbation of the symptoms [29]. In these cases, the use of products secreted by MSCs, such as the secretome or exosomes, can represent a safer strategy, avoiding the risk of pulmonary retention or vascular embolism. Another disadvantage, which is difficult to overcome, is the potential recognition and removal of the administered MSCs by the cells of the innate immune system [21].

Finally, the use of biomaterials in specific and adapted strategies are new approaches that guarantee cell survival and support their therapeutic action. The use of scaffolds allows retention and stabilization of MSCs and ensures a continuous release of their soluble factors. The use of tubular structures, for example in the treatment of peripheral nerve injuries, enables retention of cells and their secretion products in the injury site, protected from the hostile environment of the neighborhood [30,31]. The biophysical characteristics of the applied materials are an important concern in these approaches, with factors such as thickness, porosity or electroconductivity having an influence on cell survival, differentiation, and action [32]. The use of 3D scaffolds further increases the potential for therapeutic efficacy when compared to traditional 2D scaffolds, since they can maintain cell-to-cell interaction and support the extracellular matrix, stimulating the expression of specific regulatory genes important for the approach of a given clinical condition [33].

### 2.2. Homing

Regardless of the method of administration or endogenous mobilization, to reach the site to be regenerated, MSCs respond to the gradient of chemoattractants to lodge within the specific tissues [34]. Pro-inflammatory chemokines produced and released at the site of injury, such as tumor necrosis factor-alpha (TNF-α) and histamine, activate blood vessel endothelial cells, inducing the activation of P-selectins, vascular cell adhesion molecule 1 (VCAM-1) and the intercellular adhesion molecule 1 (ICAM-1) [35]. When MSCs “roll” over the surface of the vascular wall, they stimulate the expression of ligands important for extravasation, such as CD44 (homing cell adhesion molecule (HCAM)), CD49d and very late antigen 4 (VLA-4), and for extravasation to occur it is necessary to develop a contact surface on the MSCs that allows adhesion to endothelial cells (Figure 1).

The adhesion phase is based on the expression of CD44/HCAM by MSCs, a surface antigen that interacts with the selectins expressed by endothelial cells [36]. The specific selectins to which MSCs bind are still under debate. Some studies show that MSCs bind to endothelial cell P-selectins, but they do not express the P-selectin glycoprotein ligand-1 (PSGL-1). More recent works have identified galectin-1 and CD24 as potential P-selectin ligands used by MSCs at this stage [37,38]. In addition, MSCs also express platelet- and neutrophil-derived molecules such as fibroblast growth factor receptors (FGFR), which in turn interact with basic fibroblast growth factors (bFBF) present in endothelial cells and mediate the adhesion of ligands to P-selectin [39]. The firm adhesion to the endothelium is ensured through an modification in the signaling expression pattern [11].

The activation phase is related to the G protein-coupled chemokine receptors, associated with the response to inflammatory signals. The aim of this phase is to increase the affinity of integrins, essential in the entrapment stage, through the induction of conformational changes in their extracellular domain [11]. Expression of stromal cell-derived factor (SDF)-1 by endothelial cells is essential at this stage, as SDF-1 is the ligand for the chemokine receptor CXCR4, which is thought to be expressed in MSCs [40]. However, there are other works that suggest that MSCs do not express CXCR4 and point to other receptors that also bind to SDF-1 such as CXCR7 and to other chemokines such as monocyte chemoattractant protein (MCP)-1 and MCP-3 and their respective cell receptors, like Chemokine receptors (CCR)-2 [41,42,43]. Regardless of this, the expression of this receptors directly influences which tissue the MSCs will migrate into.

The entrapment phase involves the action of integrins. MSCs express VLA-4 (α4β1 integrin) whose activation occurs secondarily to SDF-1 binding to chemokine receptors. Once activated, VLA-4 binds to VCAM-1 present on the surface of endothelial cells [44]. It is important to remember that MSCs themselves express integrin ligands such as VCAM-1 [45], although their functions and importance have not yet been determined. When cells are exposed to pro-inflammatory cytokines such as interleukin-8 (IL-8), activation of phospholipase (PLC) occurs, which results in an increase in the concentration of intercellular calcium. This increase promotes downstream signaling where there is an increase in tailin, GTPases (Rho and Rap1) and guanine-exchange factors [20,46]. Some of these factors can improve the affinity between VLA-4 and VCAM-1 [47]

In the diapedesis phase, cells will undergo transmigration through the endothelial cell layer and basement membrane of the vascular endothelium. Once MSCs adhere to the endothelium, they undergo a conformational change and form filopodia, due to chemokine ligand 9 (CXCL-9) stimulation. The transmigration occurs following the secretion of matrix metalloproteinases (MMPs) which will break down the basal membrane type IV collagen [44]. The expression of these enzymes is stimulated by inflammatory cytokines produced by injured tissues [48] and is inhibited by local proteins such as tissue inhibitors of metalloproteinases (TIMPs), but the latter also has a relevant role in the maturation and activation of MMP2 from its proenzyme [49]. Further details about the transmigration process are not yet understood.

In the last phase, after crossing the basement membrane, MSCs migrate to the specific site of the lesion. Cells are attracted by chemotactic signals produced and released by the injured tissue, such as platelet-derived growth factor (PDGF), insulin-like growth factor (IGF)-1 and chemokines such as SDF-1 [50]. Exposure of cells to TNF-α increases the sensitivity of MSCs to chemokines by upregulating CCR2, CCR3, and CCR4 receptors, and IL-8 also promotes migration to sites of injury and secretion of pro-regenerative factors such as endothelial growth factor (VEGF) [51,52]. Migration is complete when MSCs reach their target tissue, where they can then exert their therapeutic, pro-regenerative and immunodulatory activity.

#### Homing Improvement Techniques

Some simple techniques can improve the effectiveness of homing and overcome some limitations associated with delivery techniques, such as the use of anticoagulants and vasodilators to reduce the occurrence of trapping in the lungs and increase homing to injured tissues [53]. However, the homing process after mobilization and migration is highly dependent on the expression of specific molecules such as CXCRs, which limits the success in cases of low expression levels. There is evidence that cells maintained in culture for long periods of time have a decreased expression of homing molecules [54]. To overcome these limitations, some alternatives have been explored to improve the homing process of MSCs.

Target administration, already described as direct administration to or near the target tissue, is a logical approach to ensure better cell retention. However, few studies have conducted a direct comparison between the homing efficacy of systemically and locally administered MSCs, and in some works where this comparison was made, better outcomes were observed after intravenous administration [55]. The use of MSCs’ sheets, that is monolayers of MSCs in culture that are detached and directly applied locally, seems to be a good alternative to traditional target administrations, with the clinical application demonstrating good results and better performance when comparing to direct MSC administration. The administration of scaffold-free MSC-sheets promoted structural and functional recovery in cases of ischemic cardiomyopathy, ensuring cell survival and myocardial recovery [56]. Transplantation of MSC sheets directly into a bone fracture also led to the formation of new bone in the fracture gap and complete bone union [57]. Thus, transplantation of MSCs’ sheets presents advantageous characteristics that justify further exploration, namely the lower risk of systemic dilution and embolism, greater cell survival indexes and increased secretion of paracrine factors [11].

The in vitro priming aims to change the culture conditions to indirectly influence the gene expression of MSCs and the different steps of homing. Simple methods such as changing coating in MSCs’ cultures with hyaluronic acid significantly improved CD44 upregulation and homing at the site of inflammation [58]. However, the main approaches are attempts to increase the expression of homing receptors by supplementing the culture medium with soluble factors such as cytokines (TNF-α, interferon gamma (IFN-γ)), interleukins, growth factors and other molecules, maintaining the culture under hypoxic conditions or even priming with biomaterials [59]. The priming of MSCs with IFN-γ and TNF-α in different assays translated in adaptations such as increased capacity to inhibit T cells through upregulation of Indoleamine 2, 3-dioxygenase (IDO) and through programmed death-ligand 1 (PD-L1), higher secretion of IL-10, greater differentiation of macrophages into anti-inflammatory phenotypes and a greater ability to inhibit the degranulation and proliferation of cytotoxic T cells even after cell thawing. Priming with interleukins such as IL-1β, on the other hand, allowed an increase in the secretion of trophic factors and extracellular matrix adhesion factors, an increase in the secretion of provasculogenic factors and an increase in the potential for migration to inflammation sites through upregulation of CXCR4. In vivo, the application of primed cells also resulted in attenuation of inflammation and improved vascularization [59]. Alternatively, priming in hypoxia and with substances that promote an increase the expression of MMPs can also translate into an increase in the migration processes [60]. MSCs conditioned at 1% oxygen levels, for instance, led to higher cell survival rates by triggering anti-apoptosis mechanisms and improved the ability to produce and secrete pro-angiogenic factors, also improving cell survival after administration in vivo. [61,62].

It has already been noticed that the level of confluence of MSCs in culture can influence homing ability, with MSCs at high confluence expressing genes related to cell activation, migration and immunomodulation capacity, and cells at low confluence expressing genes related to proliferation [63]. It remains, however, to understand how these variations influence therapeutic efficacy. The co-culture of MSCs with other cell populations also influences migration of MSCs, tending to promote an upregulation of genes associated with proliferation, homing and immunomodulation [11].

Genetic modification of MSCs to induce permanent or transient overexpression of homing factors can be achieved using viral transfection. In this technique, a pre-built gene cassette is introduced into a viral vector which in turn is introduced into MSCs. Inside the cell, the viral vector induces overexpression of specific genes that influence homing factors. Transgene expression can either be permanent, translating into a continuous synthesis of specific molecular proteins, or be regulated by a gene switch [64]. This type of gene therapy, however, in addition to its high cost and technical complexity, is associated with risks, such as the integration of viral DNA that may lead to the occurrence of insertional oncogenesis [65].

To avoid the obstacles associated with genetic manipulation, chemical modification of the cell surface is a viable alternative. In these techniques, modifications are always temporary, but they usually last long enough to guarantee transmigration a few hours after administration [66]. In a recent work, CD44, the ligand naturally expressed by MSCs, has been enzymatically modified to HCELL, the E- and L-selectin ligand used by hematopoietic stem cells for homing in on bone marrow, allowing MSCs to home in on this tissue as well [35]. This same conversion can also be achieved via genetic modification, with the associated disadvantages and risks already described [67]. Through cell surface engineering methods, it is also possible to carry out a direct conjugation of the desired ligands, namely selectin ligands [68]. It is even possible to promote an attachment of antibodies to the cell surface, ensuring better homing rates in injured tissues, better survival rates and better clinical outcomes [69]. Despite representing recent and still little explored techniques, the works in which they were applied revealed the maintenance of cell viability and cell adhesion, proliferation, and differentiation capacities. They are, however, complex and technically challenging methods [11].

From a completely opposite perspective, instead of directly modifying MSCs to improve their homing ability, it is also possible to modify the target tissue to create more attractive conditions for MSCs. This technique may involve direct injection of homing factors into target tissue prior to MSC administration (still no published works to demonstrate effectiveness), genetic modification of target tissues (with the same limitations of technical complexity and risk of insertional mutagenesis), the implantation of scaffolds capable of releasing homing factors, as well as radiotherapy and ultrasound techniques that increase the rate of MSC engraftment [11].

### 2.3. Direct Immunomodulation

MSCs have the ability to modulate the immune response in different directions depending on physiological conditions, the microenvironment where they are inserted and also on the levels of hypoxia and stimulation by inflammatory factors [70].

The immunomodulatory products produced by MSCs depend on their phenotype at the time, which in turn results from the stimulation of receptors present in the cell surface and known as Toll-like receptors (TLR), by paracrine factors circulating in the microenvironment. Thus, according to the paradigm of MSCs polarization, these cells can be divided into two phenotypes: MSC1 cells have pro-inflammatory activity and are activated through TLR4 receptors; MSC2 cells, on the other hand, have anti-inflammatory or immunosuppressive activity and are activated via TLR3 receptors [71]. Another factor that can influence the immunomodulatory action of MSCs is the hypoxic microenvironment. Compared to cells in normoxia, hypoxic-exposed cells express higher levels of FAS ligand (FasL) and IL-10, maintaining an essentially anti-inflammatory function. Likewise, these hypoxic cells promote a macrophage shift to an anti-inflammatory phenotype and are more resistant to natural killer (NK)-induced lysis. In contrast, anoxic cells increase the production and secretion of Netrin-1 and have relieving effects on the inflammatory process [59]. In both cases, after stimulation, MSCs produce and secrete the corresponding immunomodulatory factors, leading to different influences on inflammatory cells, both from the innate and adaptive immune systems [71]. Specifically, MSCs have direct influence on adaptive immune system cells (T lymphocytes and B lymphocytes), and innate immune system cells (neutrophils, macrophages, dendritic cells (DC), and NK). The interactions between the stem cells and the cells of the immune system indicate that a bilateral regulation is established between them to guarantee homeostatic immunity and inflammation regulation (Figure 2).

Crosstalk between cells is multifactorial, and communication between MSCs and cells of the immune system can take place in different ways, with cell-to-cell communication or direct signaling being the simplest method of intercellular communication. This cell-to-cell contact, however, is not always essential since indirect crosstalk communication is also possible but, in specific situations, the absence of direct signaling can limit the effective potential of MSCs. If there is no direct contact between MSCs and lymphocytes, for example, the expression of certain markers or the formation of adhesion molecules such as ICAM-1 and VCAM-1 is not observed [72,73]. Although there is evidence that without direct contact the upregulation of surface proteins capable of suppressing the inflammatory response is more difficult, namely without the activation of PD-L1 and FasL, other works indicate that the inhibition of the programmed death-1 (PD-1) in T cells can be achieved even through indirect co-culture systems of MSCs with immune cells [74,75]. Some cell types such as B lymphocytes are more efficiently immunosuppressed in direct co-culture with MSCs, and DC immunomodulation is more effective in a mixed process where both a direct cell contact and the production of soluble factors such as IL-6 occurs [76,77].

#### 2.3.1. Influence over the Adaptive Immune System

##### T-Cells

When activated by the presence of pro-inflammatory cytokines such as IFN-γ, TNF-α and interleukins such as IL-2, IL-1α and IL-1β, MSCs have a great influence on the action, survival, and migration of T lymphocytes. These factors are produced during the acute phase of inflammation by the T cells themselves or by other cells, and in sequence MSCs can either attract T lymphocytes to the vicinity to promote their inhibition or, upstream, inhibit their initial activation [70,78]. During the T cell activation phase, MSCs seem to inhibit the expression and early activation markers such as CD25 and CD69, although this mechanism seems to be inconsistent in different cell populations and in different contexts of immunoregulation [79,80]. The level of interference of MSCs in the production of cytokine secretion by activated T lymphocytes is also a matter of debate, with studies demonstrating both a decrease and an increase in IFN-γ secretion, with variations depending on the origin of the lymphocyte population [81,82].

The effects of MSCs on T cell proliferation appear to be dependent on the method of activation, and the methods of action range from a dose-dependent mechanism to an inhibition through polyclonal activators, either by cell-to-cell contact or by the paracrine route [82,83]. Secreted factors capable of promoting proliferative inhibition include, among others, IDO, (prostaglandins) PGE2, inducible nitric oxide synthase (iNOS), (cyclooxygenase 2) COX2 for the generation of nitric oxide (NO), Galectins, IL-10, Human leucocyte antigen G (HLA-G5), hepatocyte growth factor (HGF) and transforming growth factor beta 1 (TGF-β1) [80]. These factors are produced in response to T lymphocyte activation and may act as ligands for the T cell-specific chemokine receptors (42). Thus, these chemokines are able to attract T cells to their vicinity, where, for example, the catabolization of IDO produced by MSCs results in catabolites capable of inhibiting T lymphocytes and leading to their apoptosis [84]. All these factors act in an interrelated way, and, for instance, when a decrease in TGF-β levels is observed, a decrease in IDO is also detected, thus attenuating the effects of MSCs on T cells and promoting an immune response. The IDO is thus a reciprocally mediated regulatory system, varying according to the balance between stimulus, immune response and MSC intervention [85]. The HLA-G acts at different levels, promoting the activation and differentiation of regulatory T lymphocytes (Treg) and interfering with the action of cytotoxic CD8+ and CD4+ T lymphocytes. Additionally, this antibody also impairs the activation of B lymphocytes, the maturation of DCs, the activation and proliferation of NKs and, in general, attenuates the exuberant immune response in autoimmune diseases [86,87]. Galectins produced by MSCs are essential molecules in their immunosuppressive effect, and while galectin-3 can modulate the behavior of migration, adhesion and proliferation of T and B lymphocytes, galectin 1 decreases the release of inflammatory cytokines such as Interleukins, TNF-α and IFN-γ [88]. Other factors such as PD-L1 and FasL seem to act over T cells mainly through mainly direct contact mechanisms [80]. The PD-1 proteins are expressed in T cells and play an important role in regulating their activation and in homeostatic control. PD-L1, in turn, is expressed in MSCs. Through the PD-1/PD-L1pathway, by direct contact, MSCs can inhibit the activity of T lymphocytes by inhibiting glucose uptake, which has an important role in ensuring an attenuation of the activity of these cells and promoting a T-cell tolerance. As mentioned, it was traditionally thought that this inhibitory effect would be lost once the cells were physically separated [89], but recent evidence shows that it can be maintained even trough paracrine effects [74,75]. Likewise, the transmembrane protein Fas is expressed on T cells, while FasL is expressed on MSCs. Through the Fas/FasL pathway, MSCs can induce T cell apoptosis by direct contact. Additionally, MSCs can also express Fas, and through it control MCP-1 secretion, attracting T cells to the neighborhood and thus ensuring cell-to-cell contact and T cell inactivation [90]. Whether direct contact is effectively an alternative method of action for MSCs on T cells or just an essential component for effective immunoregulation is still a matter of debate. It is also important to note that MSCs do not always have an inhibitory effect on T cell proliferation. In situations where the inflammatory process is not intense enough, MSCs can adopt an MSC1 phenotype, and therefore a pro-inflammatory influence, attracting and stimulating an infiltration of T cells to exacerbate the local immune response [91,92].

Differentiation of activated helper T cells into one of the possible subtypes, Th1, Th2, Th17 or Tregs can also be modulated by MSCs. Activation of naive T cells in the presence of MSCs can lead, depending on the conditions, to differentiation into Th1 with production of INF-γ and IL-2 or into Th2 with increased production of IL-4 and IL-10 [83]. MSCs further inhibit the differentiation of CD4+ lymphocytes into Th17 while stimulating the production of IL-10 and expression of Foxp3 transcription factor associated to differentiation into Tregs [80,93].

##### B-Cells

The type of communication established between MSCs and B cells is still largely misunderstood, although it is known that the activation of MSCs interferes with the proliferation of B lymphocytes, their differentiation into plasmocytes, the expression of chemokine receptors and the production and release of immunoglobulins [80]. The inhibition of proliferation seems to be related to an interference in specific phases of the cell cycle and not to an induction of apoptosis [94]. These actions appear to occur both through a cell-to-cell interaction and through the inhibition of IFN-γ secretion by T-cells, and may require the presence of CD3+ T cells [95]. Thus, inactivated MSCs not only do not have inhibitory effects on B lymphocytes but allow their proliferation along with other inflammatory cells. When in co-culture, the presence of MSCs inhibits B cell proliferation, and their secretion products inhibit B lymphocyte maturation and induce a downregulation of several factors [96]. Among other possible mechanisms of action, MSCs can interfere with the differentiation of B lymphocytes by decreasing immunoglobulins such as IgA, IgG and IgM and altering their chemotactic characteristics and chemokine receptors [94,95]. Clinically, there is also evidence of an inhibitory effect of MSCs on B lymphocytes with therapeutic benefits in cases of systemic lupus erythematosus [13].

Alternatively, MSCs can stimulate B cell proliferation through the production of B cell activating factor, with its downregulation having the opposite effect [97]. IDO produced by MSCs has a pro-survival effect on CD5+ B cells, which in turn produce IL-10 and induce Treg differentiation [98]. Other trophic factors for B cells can be produced by MSCs, such as IL-6, IL7 or ILGF-1 [99].

#### 2.3.2. Influence over the Innate Immune System

##### Natural Killers

The Influence of MSCs on NK is accomplished at different levels, with the cells of the innate immune system being impacted on their proliferation, cytokine secretion, phenotypic characteristics, and cytotoxic effects. Considering that the action of NKs is related to the signals transmitted by receptors that interact with HLA molecules, HLA negative cells can be potential targets of these cells and MSCs can be lysed by NK [100].

There are still doubts about the method of inhibition, with some studies indicating that the process occurs in a cell-to-cell manner and others showing communication via the IDO system, PGE2 or HLA-5G [101,102]. Upon recognition of NK, MSCs inhibit its proliferation by ensuring the downregulation of activating interleukins such as IL-2 and IL-15, and a decrease in the characteristic secretion profile (IFNγ, TNFα, IL-10) and lytic activity is also observed [80,103]. The contact between MSCs and NKs inhibits the cytotoxicity of the latter in tumoral cell lineages [104], but other studies point in the opposite direction, indicating that under certain conditions MSCs can even promote NK function and expansion [105], stimulating the production and secretion of IFNγ and its cytolytic activity [106].

##### Dendritic Cells

MSCs can interfere with the maturation, recruitment, and migration of DCs, in addition to inhibiting the differentiation of monocytes into DCs. This inhibition of differentiation and maintenance of DCs in an immature state is achieved alongside an interference with the upregulation of markers such as CD80, CD83, CD86 and HLA-DR, but is generally a reversible process after removal of the MSCs influence [107,108]. However, some works show that MSCs do not interfere with the change of DCs from an immature to mature state through the action of lipopolysaccharides (LPS), yet with derivations in the results [107].

Furthermore, MSCs prevent the secretion of TNF-α from DCs activated by LPS, which in turn has an inhibitory effect on the maturation and migration of these cells to the lymph nodes, on the expression of receptors necessary for the capture and processing of antigens and consequent stimulation of T lymphocytes [83]. The secretion of interleukins by DCs, such as IL-12, can also be impaired, preventing the differentiation of T cells into Th1, guiding a differentiation into Th2 and Treg [109] or inducing a state of T cell anergy [107].

##### Macrophages

MSCs can influence the maturation, functionality, and migration of macrophages, thus influencing the immune response associated with these cells through the production of different factors. IDO expression by MSCs after interaction with inflammatory cytokines promotes the differentiation of monocytes into M2 phenotype macrophages, therefore, with anti-inflammatory and immunosuppressive functions. Induction of the conversion of M1 (pro-inflammatory) to M2 macrophage phenotypes is also possible, namely through the production of PGE2 by the MSCs [110]. PGE2 and other factors such as the IL-1 receptor antagonist promote the production of IL-10 and TGF-β by macrophages, leading to suppression of the immune function of other cells such as NK, Tregs and CD8+ and CD4+ T cells [111]. Influenced by the tumor microenvironment, and through the production of specific chemokines such as CCL-2, CCL-7 or CCL-12, MSCs are able to recruit CCR2 expressing monocytes to tumor sites, promoting the local concentration of macrophages and tumor growth. The decrease in CCL-2 influences the opposite pathway in the regulation of bloodstream monocytes migration [112]. Through the production of factors such as keratinocyte growth factor (KGF), TGF, epidermal growth factor (EGF), VEGF-α and macrophage inflammatory protein MIP, the MSCs are able to recruit both macrophages and endothelial cells to skin lesion sites, promoting wound healing [113]. The transplantation of MSCs and their action on macrophages can also have a direct influence on the clinical improvement of kidney and liver diseases and even in sepsis resolution, primarily through the promotion of IL-10 expression in macrophages [106].

##### Neutrophils

MSCs can stimulate the activity and maintain the viability of neutrophils for long periods of time, and when in co-culture, can promote the production of factors such as TGF-β, IFN-α and granulocyte colony-stimulating factor, even at a small MSC: neutrophil ratio. These effects are essentially related to the production of IL-6 [114] and other factors such as CXCL1, CXLC2, CXCL5 and CXCL8 essential for their mobilization and infiltration [115]. Having an antimicrobial function, MSCs produce and secrete antimicrobial peptides whose action on bacteria is both direct (loss of cell membrane integrity) and indirect, through the release of pro-inflammatory cytokines that attract immune cells such as neutrophils. Once established, neutrophils intervene in the acute phase of inflammation through different mechanisms, namely releasing lytic enzymes, producing reactive oxygen species and producing neutrophil extracellular traps, which together help to eliminate invading bacteria [115,116]. This antimicrobial activity can sometimes have negative effects at the systemic level, and as such MSCs can also contribute to the creation of a tolerant immune system, an immunosuppression which in turn interferes with the body’s defense mechanisms and promotes microbial proliferation [47] creating another function dependent on the balance between the action of MSCs and the systemic response [117]. In some situations the bactericidal activity of neutrophils can additionally have inappropriate effects and trigger neutrophil-mediated tissue damage, situations where MSCs can also intervene beneficially [118].

### 2.4. Alternative Immunomodulation Methods

Mitochondrial transfer is an alternative method through which MSCs are able to reprogram host cells, in a process that requires cell-to-cell contact through gap junctions, nuclear loss by cell fusion or through the development of intercellular nanotubes [119]. This transfer can have several effects on the recipient cell, namely increasing its protection against injuries by hypoxia, oxidative stress, and radiation, but also allowing the resumption of aerobic respiration after recovery of the mitochondrial membrane potential [120]. Finally, mitochondrial transfer also allows modulation of the activity of cells of the immune system [121]. Rho-GTPase Miro 1 upregulation seems to be essential for the mitochondrial transfer and the consequent therapeutic effects of MSCs to occur [122].

Phenomena related to necrobiology, that is molecular, biochemical, morphological and functional changes that occur in a tissue or cell in response to cell death in the vicinity, are also a method by which MSCs can influence other cells, with bioactive components of dead or dying cells inducing activation of immunomodulatory pathways [123]. The most effective necrobiological phenomena are apoptosis and autophagy. Apoptosis occurs in MSCs often through nutrient deprivation but also through NO action. Inhibiting NO, despite allowing the inhibition of apoptosis processes, also inhibits their immunosuppressive capacity [77]. Some studies demonstrate that apoptotic MSCs may have a greater immunosuppressive effect than their healthy counterparts [124], in addition to being more effective in attenuating organ damage in animals undergoing sepsis processes [125]. Monocytes that phagocytized apoptotic MSCs also demonstrate changes in the expression of IDO, COX2 and PD-L1 and also secrete smaller amounts of TNF-α and greater amounts of PGE2 and IL-10 [126]. Autophagy, on the other hand, is an essential mechanism in MSCs biology to ensure genomic stability, prevent cell senescence and ensure maintenance of differentiation and multipotency, but it may also play an important role in MSCs immunoregulation [127]. Autophagy in MSCs is induced by hypoxia and nutritional depletion, having a protective effect on cells in vitro [128]. Immune system cells co-cultured with autophagic MSCs are inhibited, and these cells also have protective effects on damaged organs. These effects are lost when autophagy phenomena are inhibited. [129,130]. Therapeutically, autophagic MSC cells also appear to have a greater neuroprotective effect in animal models of Parkinson’s disease [131].

### 2.5. Paracrine Immunomodulation

Advances in knowledge around MSCs have shown that after transplantation the survival rate and the proportion of cells that effectively integrate into the recipient tissue are low. This fact reveals that, in addition to their ability to influence neighboring cells and tissues through a cell-to-cell interaction, MSCs also act through a paracrine signaling mechanism [132]. In this process, once activated after contact with inflammatory cytokines and interleukins present in the inflammatory environment where they are integrated, MSCs produce and secrete a myriad of soluble molecules such as cytokines, chemokines, growth factors and extracellular vesicles that will responsively influence the inflammatory process (Figure 3).

#### 2.5.1. Secretome

The set of soluble factors secreted by a cell stimulated by certain environmental conditions is called secretome [133]. The list of factors produced and secreted by MSCs is extensive and not yet fully described, but includes elements such as IDO, SDF-1, PGE2, IL-6, TGF-β, various growth factors and chemokine ligands. Taken together, these factors can modulate the inflammatory response in different directions, stimulate the differentiation and proliferation of local progenitor cells, promote angiogenesis, and increase endothelial and fibroblastic activity [134].

Activation of MSCs through inflammatory stimuli or crosstalk with damaged cells is an essential step for their subsequent paracrine action and to produce therapeutically influencing factors and chemoattractant molecules. After tissue injury, in the acute inflammatory phase, an elevated expression of pro-inflammatory cytokines is observed, which in turn can contribute to the activation of a large number of MSCs and of an immunosuppressive response [135]. In the chronic phase, after the institution of immunosuppressive treatments or during lesion remission, there is a decrease in the production of pro-inflammatory cytokines with an increase in anti-inflammatory ones such as TGF-β, and in the absence of an active stimulus to MSCs, some factors such as chemokines and IDO may be present in concentrations lower than the immunosuppressive threshold, allowing the proliferation of T lymphocytes and the reactivation of the immune response. This cycle confirms the functional adaptation of MSCs depending on the nature of the immune stimulus received, thus exerting adapted immunotolerant or immunosuppressive paracrine activity depending on the inflammatory environment [136].

Taking advantage of the paracrine immunomodulation mechanisms of MSCs, some techniques have been used to maximize their effectiveness in a therapeutic context, namely the administration of pro-inflammatory cytokines or anti-inflammatory cytokine inhibitors in association with cells to guarantee their action [13]. However, it has also been shown that the simultaneous use of MSCs with conventional immunosuppressive drugs is a deficient approach since such drugs can inhibit the immunosuppressive effects of MSCs [137]. Additionally, it has also been demonstrated that the paracrine efficacy of MSCs can be affected both by in vitro factors such as the number of passages, culture conditions and donor origin, as well as by the in vivo administration method. The method of administration has direct influence on the ability to migrate and homing. The culture conditions, on the other hand, can affect the cell attachment capacity by influencing the pattern of the VLA-4 α4 subunits depending on the isolation and expansion methods [11,42]. Thus, an alternative method to stimulate paracrine activity is the in vitro pre-conditioning setup, with subsequent use of the conditioned medium (CM) as a therapeutic factor. It is certain that depending on the factors used during pre-conditioning, different signaling pathways will be activated. Conditioning under hypoxic conditions, in 3D cultures or with soluble factors such as SDF-1 and TGF-β seem to stimulate the production of cytoprotective molecules, pro-regenerative and pro-angiogenic factors, and immunomodulatory cytokines (IDO, PGE2, IL-6). Preconditioning of MSCs in hypoxia revealed an increase in their functional survival and their therapeutic effects on cardiac function in models of ischemia [138]. Preconditioning with inflammatory cytokines such as IFN-γ, on the other hand, activates TLRs in MSCs, thus stimulating similar factor production and cytokine secretion [139]. The induction of necrobiological phenomena such as autophagy is also possible through pre-conditioning methods, and MSCs subjected to rapamycin and nutritional depletion during conditioning not only followed the autophagic pathway but produced exosomes capable of attenuating acute kidney injuries and maintaining their stemness characteristics after exposure to reactive oxygen species (ROS) [140,141]. The complexity behind these mechanisms, however, is still not fully understood.

The secretome can act directly either through the action of growth factors or cytokines. For example, the use of secretome derived from MSCs containing a mixture of growth factors such as PDGF, brain-derived neurotrophic factor, FGF and HGF is capable of stimulating the production of new blood vessels and brain cells, while attenuating the inflammatory process and cell death in stroke scenarios [142]. The presence of IGF-1 also has protective effects in cases of brain stroke, reducing the volume of the brain infarction, controlling inflammation and ischemia, and improving cell and brain function [143]. In skin lesions, a secretome rich in VEGF, HGF, KGF TGF-β induces cell proliferation and migration and promotes a faster reduction in wound dimensions [144]. Additionally, the growth factors present in the secretome of MSCs may also influence the promotion of angiogenesis and muscle regeneration and have therapeutic effects in common diseases of premature children [77].

The cytokines are the most important in regulation of inflammatory activity and obtain potential therapeutic effects. In the secretome produced from MSCs, cytokines with both anti-inflammatory (TNF-β1, IL-10, IL-12 p70, IL-13, IL-18, IL-25, and IL-27) and pro-inflammatory (TNF-α, IFN-γ, IL-1b, IL-6, IL-8, and IL-9) function have already been identified, whose therapeutic action and influence on the cells of the immune system will vary significantly [77]. The CM of MSCs manages to intervene in the neutrophilic activity in a distinct way, depending on the inflammatory scenario. While CM is capable of reducing intracellular hydrogen peroxide in neutrophils, prolonging their lifespan and ensuring stimulation and acceleration of bacterial uptake rate, on the other hand the paracrine crosstalk between the two cell types can be used by cancer cells to ensure tumoral growth [145]. Knowing that MSCs can be a cytolytic target of NKs, it is interesting to note that priming these cells with IFN-γ promotes an increase in the surface expression of HLA-II molecules, which has a protective effect against their cytolytic effect [100]. Paracrine factors indirectly influence T cells through interaction with cells of the innate immune system such as macrophages or DCs. Since the factors secreted by MSCs can influence the antigen-presenting cell ligands, and these are essential for activating T cell receptors, the use of the secretome can impair the entire process of activating T lymphocytes. In addition, the MSCs secretome can also regulate the expression of interleukins such as IL-1, IL-10 and IL-12 and TGF-β in antigen-presenting cells, indirectly interfering with the differentiation of T cells [13]. In the case of B cells, the secretome of MSCs is able to interfere with maturation through downregulation of IgG, IgM and CD138 [96]. Cytokines present in the secretome of MSCs can also have observable therapeutic effects in different clinical settings. The presence of anti-inflammatory cytokines such as IL-4 and IL-10 prevented apoptosis and destruction of pancreatic cells in a case of type I diabetes mellitus influenced by inflammatory cytokines such as IFN-γ, TNF-β and IL-1b [146], and IL-10 suppressed an inflammation secondary to transplant arteriosclerosis [147]. Likewise, cytokines such as CCL2, CCL5, CCL 7 and TNF-β, with important functions in promoting angiogenesis, osteogenesis and cell recruitment and proliferation, may have beneficial effects in bone healing [148].

#### 2.5.2. Extracellular Vesicles

Extracellular vesicles (EVs) are components generated by cells when exposed to activation or stress and that carry RNA, proteins, and lipids [149]. EVs can be divided into ectosomes (microparticles or microvesicles with submicron-size) with dimensions between 100 and 1000 nm, and exosomes (nano-size particle) with <100 nm. While ectosomes protrude from the cytoplasmic membrane in mostly physiological or pathophysiological situations, exosomes originate from multivesicular bodies secreted by exocytosis after fusion with the cell membrane, sometimes following endoplasmic reticulum stress [150]. Since EVs are surrounded by a phospholipid bilayer, they have several advantages over cells per se: low immunogenicity, stability in vivo and during freezing and thawing (facilitating banking and off the shelf application), ease of delivery and organic distribution and lack of tumorigenic activity. However, their mechanisms of action are not fully understood, and the biodistribution can be influenced by the route of administration [151].

Recent studies around MSCs’ EVs have revealed that their vesicular cargo includes the presence of different types of RNA (messenger and transfer RNA, microRNA (miRNA)), different types of proteins and peptides, enzymes, immunomodulatory factors, growth factors and even lipids capable of mediating cell-to-cell communication. Thus, these vesicles can therapeutically influence injured tissues, attenuating the activation of inflammatory cells and decreasing the level of apoptosis and oxidative stress [152]. Examples of proteins found inside EVs secreted by MSCs include osteoprotegerin and angionenin, which play an important role in bone regeneration [148].

EVs can act directly on the innate and adaptive immune response, even managing to influence the progression of autoimmune diseases. Exosomes extracted vary according to previous stimulation, and are able to inhibit the production and secretion of pro-inflammatory factors such as TNF-β, IL-1β and IL-6 by immune cells while increasing the concentration of anti-inflammatory factors such as TGF-β [152,153]. CCR2 positive exosomes released by MSCs may interfere with macrophage activation, altering CCL2 recruitment mechanisms, and may also inhibit TLR4 receptors on these cells [154,155]. The effect on DCs occur in both mature and immature cells, capable of engulfing EVs from MSCs and thus being induced to produce IL-10. Additionally, these pre-treated DCs can later influence other cells such as T lymphocytes, promoting a decrease in the levels of TGF-β, IL-6 and IL-10 [156].

EV-mediated mitochondrial transfer, another possible mechanism, may stimulate increased phagocytic activity in cells of the innate immune system, namely macrophages, increasing the expression of the M2 phenotype [157]. Apoptotic cells, on the other hand, can produce different types of EVs and apoptotic bodies with immunomodulatory influence, the latter being rich in spliceosomes capable of altering RNA splicing in recipient cells [158]. EVs derived from apoptotic bodies have also demonstrated the ability to influence inflammatory processes, infections, immunomodulation and autoimmunity, being important mediators between dead or dying cells and active ones [159].

Alternatively, EVs, especially exosomes, can also act as RNA transferors between cells, thus regulating gene expression and indirectly and therapeutically influencing different clinical scenarios [13]. The miRNA present in MSCs EVs can regulate macrophage activity, controlling the production of paracrine factors. Macrophages, in turn, can also promote a decrease in the expression of some types of miRNA in MSCs, a downregulation that guarantees the secretion of anti-inflammatory factors with potential therapeutic effects [160]. Therapeutically, miRNA also has beneficial effects on the regeneration of different tissues, with miR27a, miR196a and miR206 accelerating bone regeneration [148], miR133 stimulating neuronal tissue remodeling after stroke [161] and miR125b-5p with anti-apoptotic effects in cases of myocardial infarction [138].

## 3. MSCs, Immunomodulatory Activity and Clinical Potential

As the mechanisms of action and immunomodulation of MSCs are being unveiled, a whole new range of potential clinical applications of these cells is also being established. Although MSCs are being studied and explored in virtually every field of health and medical research, autoimmune diseases have characteristics that make them especially important targets in these new therapeutic approaches. Autoimmune diseases are chronic diseases and systemic disorders in which an overactivation of the immune system occurs and a chronic inflammatory environment is established, leading to damage and general organic changes. Due to their ability to adapt to inflammatory microenvironmental conditions and to exert the immunomodulatory activity both directly and through a paracrine route, MSCs are excellent candidates to be applied in these treatments. The main mechanisms through which MSCs can positively influence the autoimmune process include the inhibition of B and T cells, the decrease in Th1/Th2, Th17/Treg, and M1/M2 ratios and the downregulation of inflammatory cytokines and factors such as IL-1, IL-6, IL-17, TNF-α, IFN-γ and the upregulation of others such as IL-4, IL-10 and TGF-β [162].

Considering the COVID-19 pandemic that has affected the world in recent years, MSCs have also been explored as experimental treatments in patients with severe acute respiratory syndrome, revealing promising results. One of the great difficulties in establishing an effective strategy in these patients is the ability of the virus to stimulate a severe storm of cytokines in the lungs, a mechanism against which MSCs have advantageous effects as they manage to establish an immunomodulated environment by secreting several immunodulatory factors with a synergistic effect on different cytokines simultaneously [163]. When administered, MSCs migrate to the pulmonary microvasculature and extravasate to the pulmonary alveoli, where they contact with a pro-inflammatory environment caused by virus replication and the consequent cytokine storm. Stimulated by surrounding cytokines, MSCs respond through both cell-to-cell contact and through release of paracrine factors with inflammatory and antimicrobial effects. Among a myriad of effects still poorly understood, based on prior knowledge, it is believed that in these cases MSCs can modulate the activation and proliferation of T cells and induce the proliferation of mononuclear cells with an anti-inflammatory phenotype. MSCs are also capable of increasing Tregs and Th2 proliferation and action, of suppressing cytokine production by T lymphocytes, of suppressing the action of NKs and DCs maturation, of inducing macrophage polarization into an M2 phenotype and of attenuating the severe inflammatory response in general. The various clinical COVID-19 clinical trials in which MSCs were applied allowed confirm a reduction in the inflammatory response and in the harmful organic effects of the cytokine storm without unexpected side effects, establishing this cell-based therapy as a promise in complicated respiratory syndromes [164].

## 4. Conclusions

MSCs have been regarded with special enthusiasm due to their multipotency and promising potential to be used in allogeneic cell-based therapies. One of the great advantages associated with their use is the ability to interact with both the innate and the adaptive immune system, with an immunodulatory influence profile. The action over the immune system occurs both through direct cell-to-cell contact and through an increasingly evident trophic and paracrine activity. MSCs’ immunosuppression potential is dependent on the sum of all chemokines, growth factors and secreted cytokines and on the intensity of the inflammatory stimulus. Furthermore, the ability to influence the inflammatory response in a particular tissue or lesion also depends on the ability of the cells to migrate to that location and on the homing process, a complex sequence of events that can be influenced externally through priming and through the method of cell administration. MSCs can be manipulated in vitro through different cell engineering techniques to modify and influence their action on different immune cells and their effectiveness during the homing process, thus creating more targeted and specific therapies for each clinical and lesion scenario.

The known disadvantages associated with the direct use of cells, such as excessive interference with organic immunity, tumorigenic potential and ectopic differentiation, as well as the evidence that paracrine activity plays a key role in the therapeutic activity of MSCs, has led to further studies on the immunomodulatory potential of their secretion products such as secretome and EVs, as well as on the influence of cell culture manipulation and preconditioning in the creation of targeted and tailored therapeutic approaches.

The immunoregulatory influence that MSCs and their secretion products exert at the organic level may allow the establishment of new therapies for the treatment of inflammatory and autoimmune diseases and for general tissue repair. Although the entire mechanism of action is complex, requires a methodical sequence of several physiological steps, and is far from being fully elucidated, the multiple studies around this topic reveal key details and have allowed for significant advances in the subject of immunomodulation by MSCs and techniques to maximize their effectiveness. The continuing commitment to explore the molecular mechanisms of MSCs and the slow but consistent transition to in vivo pre-clinical and clinical trials could lift the veil around the MSCs immunomodulatory mechanisms of action and may allow creation of new therapeutic strategies to address common medical challenges that even today have no adequate clinical approaches.

## Figures and Tables

**Figure 1 pharmaceutics-14-00381-f001:**
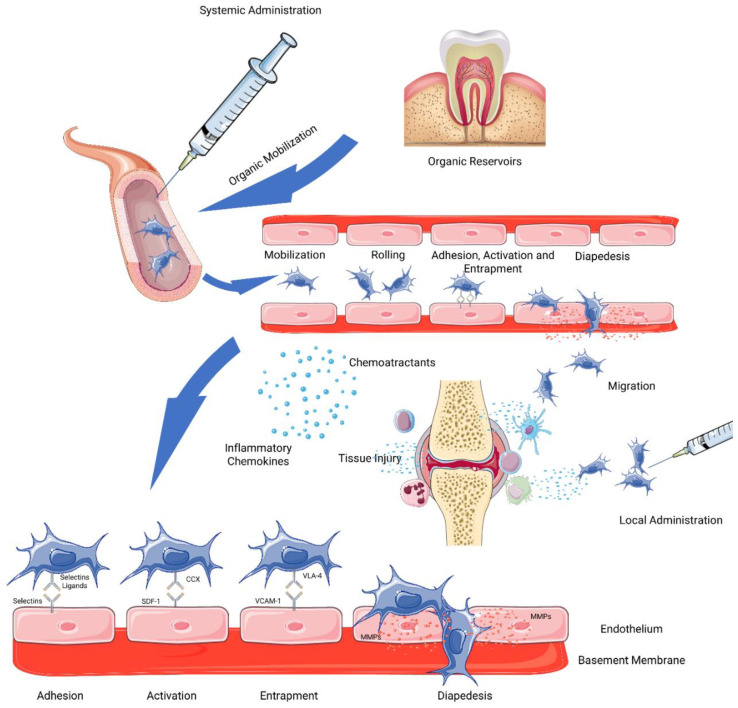
Schematic representation of the Migration and Homing processes, secondary to systemic administration, local administration, or organic mobilization of MSCs. (Adapted with permission from Servier Medical Art, licensed under a Creative Common Attribution 3.0 Generic Licence. https://smart.servier.com/ accessed on 5 January 2022).

**Figure 2 pharmaceutics-14-00381-f002:**
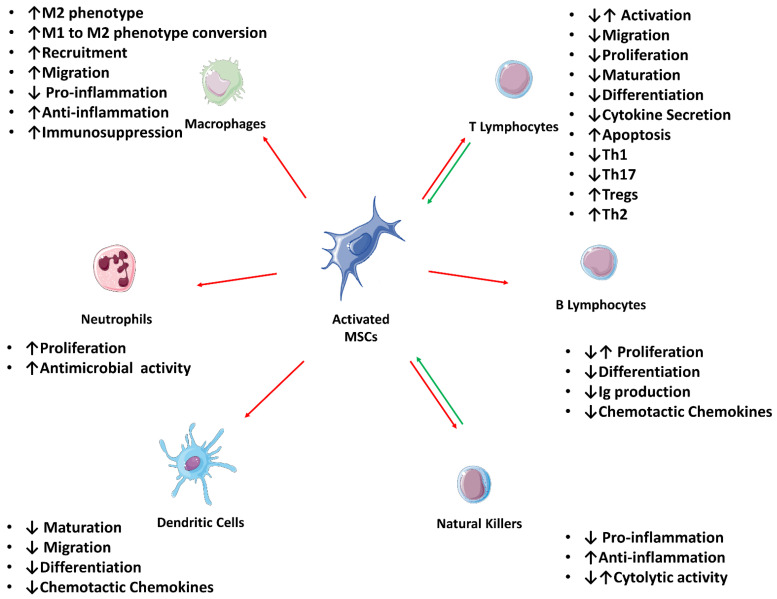
Functional and immunomodulatory influences of MSCs on cells of the Innate and Adaptive Immune System. Red Arrows—Action of MSCs on Immune System cells. Green Arrows—Action of Immune System Cells on MSCs, with activation and retroactive stimulation. (Adapted with permission from Servier Medical Art, licensed under a Creative Common Attribution 3.0 Generic Licence. https://smart.servier.com/ accessed on 5 January 2022).

**Figure 3 pharmaceutics-14-00381-f003:**
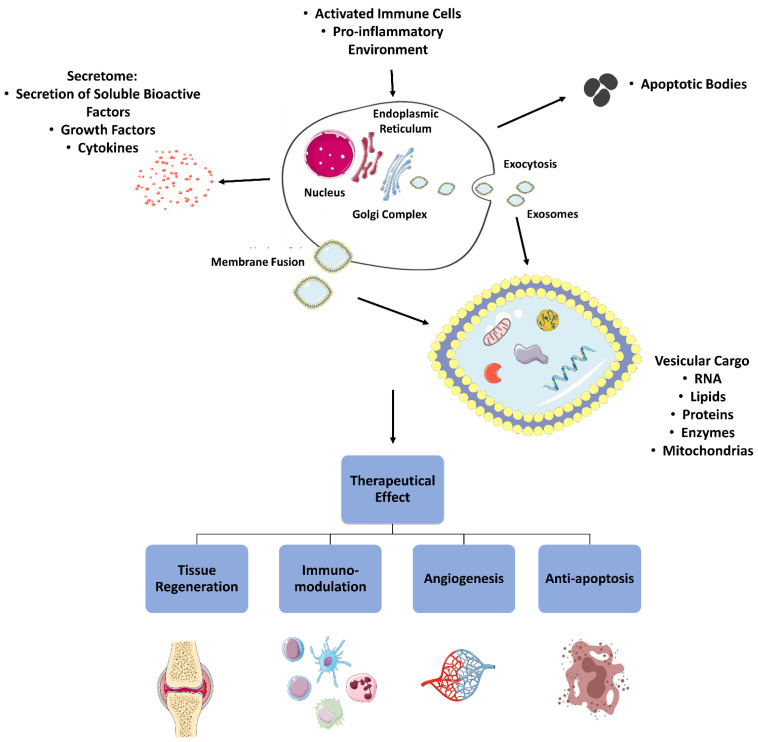
Schematic representation of the MSCs paracrine action mechanisms and corresponding therapeutic and immunomodulatory consequences. (Adapted with permission from Servier Medical Art, licensed under a Creative Common Attribution 3.0 Generic Licence. https://smart.servier.com/ accessed on 5 January 2022).

**Table 1 pharmaceutics-14-00381-t001:** Positive and negative markers proposed by the ISCT for MSCs identification. Adapted from [10].

**Positive Markers**	**Biological Meaning**
CD73	Production of extracellular adenosine
CD90	Cell-to-cell and cell-to-extracellular matrix interactions
CD105	Vascular hemostasis
**Negative Markers**	**Cell Exclusion**
CD11b	Monocytes
CD14	Macrophages
CD19 and CD79	B Cells
CD34	Hematopoietic and Endothelial Cells
CD45	Leucocytes
HLA- DR	Antigen-presenting cells and Lymphocytes

## Data Availability

Not applicable.

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
