# Peer review of "Mesenchymal Stem/Stromal Cells and Their Paracrine Activity—Immunomodulation Mechanisms and How to Influence the Therapeutic Potential"

_pharmaceutics, 2022, doi:10.3390/pharmaceutics14020381_

Round 1

Reviewer 1 Report

The manuscript authored by Alvites et al. aims to review the current knowledge about the immunoregulatory/immunomodulatory properties of MSCs, with a further focus on their paracrine activity and the immunomdulatory roles of their secretion products. Also, the authors enriched their review article with a revision of the experimental approaches aimed to improve MSCs migration and homing. 

The review article holds interest to the field and might be improved by addressing some issues, as follows:

  • paragraph 2.1: further literature citations might be included as support of multiple animal studies involving the direct injection of MSCs into injured tissues, i.e. muscle, bone, peripheral nerve injuries. 
  • Figure 1 reports a schematic representation of a tooth and tooth-associated tissues as "organic reservoirs" of MSCs, although authors did not cite the different tissue sources from which MSCs can be derived, except for bone marrow. Please, add some proper references through the text and edit the figure accordingly. 
  • When talking about the use of MSCs cell sheets as a more effective form of cell administration further details of the mentioned studies should be given and properly cited. 
  • Paragraph 2.2.1 "Homing improvement techniques": this section provides novelty to this review article, therefore I would suggest to give further impact to it by implementing the details of the mentioned findings. 
  • Paragraph 2.3 "Direct immunomodulation". 1) Page 9, lines 312-314: this sentence should be rephrased since some recent findings have highlighted that PD-L1 can be unregulated by MSCs even after indirect co-culture system with immune cells. See for reference: doi: 10.3389/fimmu.2020.00054; doi: 10.1186/s13287-021-02664-4.  Please, also correct "Fas" to "FasL". 2) Page 10, mitochondrial transfer and autophagy: I would suggest to separate these sub-sections from the main paragraph since they cannot be ascribed to direct immunomodulation properties exerted by MSCs. 3) Page 10, lines 331-333 are in contrast with lines 351-353: the first sentence should be rephrased since the exposure of MSCs to IFN-γ and TNF-α does not induce Fas-mediated apoptosis of MSCs, indeed it triggers the activation of immunomodulatory pathways that allow stem cells to survive, keep proliferating and modulating immune cells. See for further references the following studies: doi: 10.3389/fimmu.2020.00054; doi: 10.1186/s13287-021-02664-4.; doi: 10.1016/j.stem.2012.03.007; doi: 10.1634/stemcells.2005-0008. 
  • Paragraph 2.3.1.1. "T cells": the role of the immunomodulatory Fas/FasL and PD1/PD-L1 pathways would need to be further described in this section according with the related references. Lines 389-392: this sentence refers to the ability of MSCs to switch their phenotype between pro-inflammatory and anti-inflammatory features based on the surrounding microenvironment and inflammatory stimuli. This point is interestingly reviewed here: 10.1016/j.stem.2013.09.006. 
  • References n. 40 to n. 58 are missing in references section.
  • Paragraph "paracrine immunomodulation", subsection "secretome", lines 544-546: this sentence might need to be rephrased, since so far it is well known that MSCs need to be exposed to certain stimuli to exert their immunomodulatory/immunoregulatory properties and that, according to low or high inflammatory surrounding microenvironment they might behave as either immune-tolerant or immune-suppressive, as stated above. I think that a brief revision of such properties, also in light of MSCs and MSCs-derived products pre-clinical studies concerning the treatment of autoimmune disorders and COVID-19, might provide further impact to the manuscript. 

Author Response

Answer to reviewer 1

Dear reviewer 1:

Thank you very much for the feedback on this review phase, and also for the suggestions made, which received the best attention from us. The authors inform that the final document has been revised and all suggestions made by the reviewers have been introduced, being duly identified in the final document with highlight in different colors. This review also made it possible to identify and correct some errors and typos as well as improve the general level of English.

The changes made to the document are described below. All changes introduced and highlighted text segments appear in the final document highlighted in blue.

  • paragraph 2.1: further literature citations might be included as support of multiple animal studies involving the direct injection of MSCs into injured tissues, i.e. muscle, bone, peripheral nerve injuries.

The indicated section has been improved with the addition of new references for enrichment, as proposed. The new section is transcribed below:

“Examples of direct administration with proven effects include intrathecal or intraspinal administrations for spinal cord injuries [22], intra-articular for osteoarthritis and cartilage repair [23, 24], intramuscular for volumetric skeletal muscle injuries [25], direct transplantation for resolution of peripheral nerve injuries [26] and local administration for healing of bone fractures [27].”

  • Figure 1 reports a schematic representation of a tooth and tooth-associated tissues as "organic reservoirs" of MSCs, although authors did not cite the different tissue sources from which MSCs can be derived, except for bone marrow. Please, add some proper references through the text and edit the figure accordingly.

The following passage was introduced to refer to examples of some niches where MSCs have already been identified and from where they can be mobilized, but emphasizing that there are more options in constant update. In this way, the authors consider that image 3 will no longer raise doubts:

“Dental pulp, umbilical cord and placental tissues, neonatal tissues, adipose tissue, peripheral blood, skin, and olfactory mucosa are among the most widely studied niches where MSCs can be isolated and mobilized from, but the list is long and is constantly being up-dated [4].”

  • When talking about the use of MSCs cell sheets as a more effective form of cell administration further details of the mentioned studies should be given and properly cited.

The following passage has been added, introducing more details regarding two works in which MSCs sheets were used in the promotion of cardiac and bone regeneration and respective references.

“The administration of scaffold-free MSC-sheets promoted structural and functional recovery in cases of ischemic cardiomyopathy, ensuring cell survival and myocardial recovery [56]. Transplantation of MSC sheets directly into a bone fracture also led to the formation of new bone in the fracture gap and complete bone union [57].”

  • Paragraph 2.2.1 "Homing improvement techniques": this section provides novelty to this review article, therefore I would suggest to give further impact to it by implementing the details of the mentioned findings.

The following passages have been added, giving specific examples of functional and therapeutic improvements identified in MSCs after the priming and conditioning processes:

“The priming of MSCs with IFN-γ and TNF-α in different assays translated in adaptations such as increased capacity to inhibit T cells through upregulation of Indoleamine 2, 3-dioxygenase (IDO) and through programmed death-ligand 1 (PD-L1), higher secretion of IL-10, greater differentiation of macrophages into anti-inflammatory phenotypes and a greater ability to inhibit the degranulation and proliferation of cytotoxic T cells even after cell thawing. Priming with interleukins such as IL-1β, on the other hand, allowed an in-crease in the secretion of trophic factors and extracellular matrix adhesion factors, an in-crease in the secretion of provasculogenic factors and an increase in the potential for migration to inflammation sites through upregulation of CXCR4. In vivo, the application of primed cells also resulted in attenuation of inflammation and improved vascularization [59].”

“MSCs conditioned at 1% oxygen levels, for instance, led to higher cell survival rates by triggering anti-apoptosis mechanisms and improved the ability to produce and secrete pro-angiogenic factors, also improving cell survival after administration in vivo. [61, 62].”

  • Paragraph 2.3 "Direct immunomodulation". 1) Page 9, lines 312-314: this sentence should be rephrased since some recent findings have highlighted that PD-L1 can be unregulated by MSCs even after indirect co-culture system with immune cells. See for reference: doi: 10.3389/fimmu.2020.00054; doi: 10.1186/s13287-021-02664-4. Please, also correct "Fas" to "FasL".

The passage has been rephrased to introduce the indicated information, as well as the respective references. The new passage is transcribed below.

“Although there is evidence that without direct contact the upregulation of surface proteins capable of suppressing the inflammatory response is more difficult, namely without the activation of PD-L1 and FasL, other works indicate that the inhibition of the programmed death-1 (PD-1) in T cells can be achieved even through indirect co-culture systems of MSCs with immune cells [74, 75].”

 The term "Fas" has been changed to "FasL", with its introduction in the list of abbreviations.

  • Page 10, mitochondrial transfer and autophagy: I would suggest to separate these sub-sections from the main paragraph since they cannot be ascribed to direct immunomodulation properties exerted by MSCs.

A new subsection was introduced, where mitochondrial transfer and necrobiology methods were included.

“2.4. Alternative Immunomodulation Methods”

  • Page 10, lines 331-333 are in contrast with lines 351-353: the first sentence should be rephrased since the exposure of MSCs to IFN-γ and TNF-α does not induce Fas-mediated apoptosis of MSCs, indeed it triggers the activation of immunomodulatory pathways that allow stem cells to survive, keep proliferating and modulating immune cells. See for further references the following studies: doi: 10.3389/fimmu.2020.00054; doi: 10.1186/s13287-021-02664-4.; doi: 10.1016/j.stem.2012.03.007; doi: 10.1634/stemcells.2005-0008.

The authors agree that the two passages were contradictory. The first has been modified. Below is the new version:

“Apoptosis occurs in MSCs often through nutrient deprivation but also through nitric oxide (NO) action.”

  • Paragraph 2.3.1.1. "T cells": the role of the immunomodulatory Fas/FasL and PD1/PD-L1 pathways would need to be further described in this section according with the related references.

The following passages were introduced in the final document:

“Other factors such as PD-L1 and FasL seem to act over T cells mainly through mainly direct contact mechanisms [80]. The PD-1 proteins are expressed in T cells and play an important role in regulating their activation and in homeostatic control. PD-L1, in turn, is expressed in MSCs. Through the PD-1/PD-L1pathway, by direct contact, MSCs can inhibit the activity of T lymphocytes by inhibiting glucose uptake, which has an important role in ensuring an attenuation of the activity of these cells and promoting a T-cell tolerance. As mentioned, it was traditionally thought that this inhibitory effect would be lost once the cells were physically separated [89], but recent evidence shows that it can be maintained even trough paracrine effects [74, 75]. Likewise, the transmembrane protein Fas is ex-pressed on T cells, while FasL is expressed on MSCs. Through the Fas/FasL pathway, MSCs can induce T cell apoptosis by direct contact. Additionally, MSCs can also express Fas, and through it control MCP-1 secretion, attracting T cells to the neighbourhood and thus ensuring cell-cell contact and  T cell inactivation [90].”

  • Lines 389-392: this sentence refers to the ability of MSCs to switch their phenotype between pro-inflammatory and anti-inflammatory features based on the surrounding microenvironment and inflammatory stimuli. This point is interestingly reviewed here: 10.1016/j.stem.2013.09.006.

The sentence was slightly adapted, and the proposed reference was introduced. The new sentence is transcribed below. The authors further indicate that the switching mechanisms of MSCs between an anti- and pro-inflammatory influence were also described in earlier paragraphs in the document and have been improved, as indicated in one of the review points below.

“In situations where the inflammatory process is not intense enough, MSCs  can adopt an MSC1 phenotype, and therefore a pro-inflammatory influence, attracting and stimulating  an infiltration of T cells to exacerbate the local immune response [91, 92].”

  • References n. 40 to n. 58 are missing in references section.

The authors confirmed that references 40-58 appear both in the document and in the references section. Since in the previous version of the document these references all appeared on the same page, probably by mistake the page in question may not have been sent to the reviewer.

  • Paragraph "paracrine immunomodulation", subsection "secretome", lines 544-546: this sentence might need to be rephrased, since so far it is well known that MSCs need to be exposed to certain stimuli to exert their immunomodulatory/immunoregulatory properties and that, according to low or high inflammatory surrounding microenvironment they might behave as either immune-tolerant or immune-suppressive, as stated above.

The sentence was adapted, and it is transcribed below.

“This cycle confirms the functional adaptation of MSCs depending on the nature of the immune stimulus received, thus exerting adapted immunotolerant or immunosuppressive paracrine activity depending on the inflammatory environment [136].”

  • I think that a brief revision of such properties, also in light of MSCs and MSCs-derived products pre-clinical studies concerning the treatment of autoimmune disorders and COVID-19, might provide further impact to the manuscript.

New passages were introduced to explain the switch of MSCs between the MSC1 and MSC2 phenotype through the influence of the inflammatory microenvironment and of hypoxia. The new passage is transcribed below:

“The immunomodulatory products produced by MSCs depend on their phenotype at the time, which in turn results from the stimulation of receptors present in the cell surface and known as Toll-like receptors (TLR), by paracrine factors circulating in the microenvironment. Thus, according to the paradigm of MSCs polarization, these cells can be divided into two phenotypes: MSC1 cells have pro-inflammatory activity and are activated through TLR4 receptors; MSC2 cells, on the other hand, have anti-inflammatory or immunosuppressive activity and are activated via TLR3 receptors [71]. Another factor that can influence the immunomodulatory action of MSCs is the hypoxic microenvironment. Compared to cells in normoxia, hypoxic-exposed cells express higher levels of FAS ligand (FasL) and IL-10, maintaining an essentially anti-inflammatory function. Likewise, these hypoxic cells promote a macrophage shift to an anti-inflammatory phenotype and are more resistant to natural killer (NK)-induced lysis. In contrast, anoxic cells increase the production and secretion of Netrin-1 and have relieving effects on the inflammatory process [59].”

A new sub-section called "3. MSCs, Immunomodulatory Activity and Clinical Potential" was also introduced, where, as proposed, a brief reference is made to the effects of MSCs on autoimmune diseases and COVID-19. However, as the authors consider that this section already deviates a little from the original proposal of the article, so just a brief description was introduced.

“As the mechanisms of action and immunomodulation of MSCs are being unveiled, a whole new range of potential clinical applications of these cells is also being established. Although MSCs are being studied and explored in virtually every field of health and medical research, autoimmune diseases have characteristics that make them especially important targets in these new therapeutic approaches. Autoimmune diseases are chronic diseases and systemic disorders in which an overactivation of the immune system occurs and a chronic inflammatory environment is established, leading to damage and general organic changes. Due to their ability to adapt to inflammatory microenvironmental conditions and to exert the immunomodulatory activity both directly and through a paracrine route, MSCs are excellent candidates to be applied in these treatments. The main mechanisms through which MSCs can positively influence the autoimmune process include the inhibition of B and T cells, the decrease of Th1/Th2, Th17/Treg, and M1/M2 ratios and the downregulation of inflammatory cytokines and factors such as IL-1, IL-6, IL-17, TNF-α, IFN-γ and the upregulation of others such as IL-4, IL-10 and TGF-β [162].

                Considering the COVID-19 pandemic that has affected the world in recent years, MSCs have also been explored as experimental treatments in patients with severe acute respiratory syndrome, revealing promising results. One of the great difficulties in establishing an effective strategy in these patients is the ability of the virus to stimulate a severe storm of cytokines in the lungs, a mechanism against which MSCs have advantageous effects  as they manage to establish an immunomodulated environment by secreting several immunodulatory factors with a synergistic effect on different cytokines simultaneously [163].  When administered, MSCs migrate to the pulmonary microvasculature and extravasate to the pulmonary alveoli, where they contact with a pro-inflammatory environment caused by virus replication and the consequent cytokine storm. Stimulated by surrounding cytokines, MSCs respond through both cell-cell contact and through release of paracrine factors with inflammatory and antimicrobial effects. Among a myriad of effects still poorly understood, based on prior knowledge, it is believed that in these cases MSCs can modulate the activation and proliferation of T cells and induce the proliferation of mono-nuclear cells with an anti-inflammatory phenotype. MSCs are also capable of increasing Tregs and Th2 proliferation and action, of suppressing cytokine production by T lymphocytes, of suppressing the action of NKs and DCs maturation, of inducing macrophage polarization into an M2 phenotype and of attenuate the severe inflammatory response in general. The various clinical COVID-19 clinical trials in which MSCs were applied al-lowed to confirm a reduction in the inflammatory response and in the harmful organic effects of the cytokine storm without unexpected side effects, establishing this cell-based therapy as a promise in complicated respiratory syndromes [164].”

Reviewer 2 Report

The paper is well written and addresses one of the main issues regarding MSCs, the clinical application to patients by reviewing the potential effects of MSCs in the cell therapy domain. I mean, the paper does not bring new and unreported evidences but has the merit to summarize the multiple reports in this highly expanding field. Writing a review is always a hard job and in this case I think it was worth.

Author Response

Answer to reviewer 2

Dear reviewer 2:

Thank you very much for the feedback on this review phase, for the acceptance of the article and for the compliments to the review prepared by the authors. The authors inform that the final document has been revised and all suggestions made by the reviewers have been introduced, being duly identified in the final document with highlight in different colors. This review also made it possible to identify and correct some errors and typos as well as improve the general level of English.

Reviewer 3 Report

This is good comprehensive review presenting the current state of research on immunomodulatory and immunoregulatory ability of MSCs for potential clinical use based on the biological characteristics of the MSC. In this review the Authors introduced biological features of MSCs in the context for potential clinical application and the need to meet the criteria of a biological medicinal product based on the EMA and FDA regulations.

Immunomodulatory capacity of MSCs on the innate and adaptive immune cells, paracrine immunomodulation and MSCs secretome are well summarized and illustrated.

I suggest minor corrections in term of:

  • The quality of description of Figure 3 need to be corrected by changing the size and font of lettering.
  • Some typos need corrections e.g. Line 23 “MSCS” change for “MSCs”; line 26 “essays” should be “assays”; line 282 “The Immunomodulatory…” change for “The immunomodulatory…” and few others within the manuscript body.

Author Response

Answer to reviewer 3

Dear reviewer 3:

Thank you very much for the feedback on this review phase, and also for the suggestions made, which received the best attention from us. The authors inform that the final document has been revised and all suggestions made by the reviewers have been introduced, being duly identified in the final document with highlight in different colors. This review also made it possible to identify and correct some errors and typos as well as improve the general level of English.

The changes made to the document are described below. All changes introduced and highlighted text segments appear in the final document highlighted in yellow.

  • The quality of description of Figure 3 need to be corrected by changing the size and font of lettering.

Image 3 has been modified and the font size increased for better reading. The new image was introduced in the final article and is also below for review.

  • Some typos need corrections e.g. Line 23 “MSCS” change for “MSCs”; line 26 “essays” should be “assays”; line 282 “The Immunomodulatory…” change for “The immunomodulatory…” and few others within the manuscript body.

The indicated Typos have been corrected in the new version of the article. The authors also inform that the article has been re-read and some additional typos identified have been corrected.

Round 2

Reviewer 1 Report

The authors revised and improved their manuscript according to the raised comments and suggestions.